# Correlation of Autophagosome Formation with Degradation and Endocytosis Arabidopsis Regulator of G-Protein Signaling (RGS1) through ATG8a

**DOI:** 10.3390/ijms20174190

**Published:** 2019-08-27

**Authors:** Yue Jiao, Miroslav Srba, Jingchun Wang, Wenli Chen

**Affiliations:** 1MOE Key Laboratory of Laser Life Science & Institute of Laser Life Science, South China Normal University, Guangzhou 510631, China; 2College of Biophotonics, South China Normal University, Guangzhou 510631, China; 3Department of Experimental Plant Biology, Faculty of Science, Charles University, 12844 Prague, Czech Republic

**Keywords:** *Arabidopsis*, regulator of G signaling protein 1, autophagy, nutrient starvation, glucose, BY-2

## Abstract

Damaged or unwanted cellular proteins are degraded by either autophagy or the ubiquitin/proteasome pathway. In *Arabidopsis thaliana,* sensing of D-glucose is achieved by the heterotrimeric G protein complex and regulator of G-protein signaling 1 (AtRGS1). Here, we showed that starvation increases proteasome-independent AtRGS1 degradation, and it is correlated with increased autophagic flux. RGS1 promoted the production of autophagosomes and autophagic flux; RGS1-yellow fluorescent protein (YFP) was surrounded by vacuolar dye FM4-64 (red fluorescence). RGS1 and autophagosomes co-localized in the root cells of *Arabidopsis* and BY-2 cells. We demonstrated that the autophagosome marker ATG8a interacts with AtRGS1 and its shorter form with truncation of the seven transmembrane and RGS1 domains *in planta*. Altogether, our data indicated the correlation of autophagosome formation with degradation and endocytosis of AtRGS1 through ATG8a.

## 1. Introduction

Degradation of proteins in eukaryotic cells is mediated either through autophagy or the ubiquitin proteasome system (UPS). The UPS clears short-lived proteins, whereas autophagy degrades whole organelles and individual proteins that cannot be processed by the UPS [1]. Autophagy is a ubiquitous and highly conserved process, in which degrading misfolded and long-lived proteins, and damaged or old organelles are engulfed with a double membrane and then infused to the vesicle for lysosomal degradation, (or vacuole in plant cell). Stresses including starvation and oxidation, or stimulation by glucose and hormones accelerate autophagy processes [2,3,4,5]. Autophagic activity is low and balanced when cells contain sufficient nutrients, but increases rapidly when cells are starved. Cells survive starvation by recycling fatty acids and amino acids to meet the energy demand, and remove misfolded proteins and abnormal organelles [6]. The basic autophagy process is conserved among eukaryotes from yeast to animals and plants [7,8,9]. Several types of autophagy, including microautophagy [10], macroautophagy [9], chaperone-mediated autophagy [11], and organelle-specific autophagy, have been described in species [12]. Cell autophagy is normally maintained at a relatively low level, but increases abruptly when facing a disturbance. Autophagy preserves cell homeostasis and is one of the crucial safeguards of eukaryotic organisms.

The best indicator for autophagy is the formation of autophagosomes or double membrane vesicles that sequester other organelles or proteins. As the UPS and autophagy are both vital for cellular homeostasis (amino acids and fatty acid recycling, ATP economy), their activities are accurately orchestrated and regulated [1]. Improper autophagy is associated with pathologies, such as altered cell growth and cell death [12,13]. Sugars are common sources of energy and carbon supply [14]. Glucose functions as both a nutrient and a potent signaling molecule for controlling growth and development in eukaryotes [14,15]. Glucose signaling in *Arabidopsis* is integrated with pathways including the hexokinase 1 and G-protein signaling and glycolysis-dependent pathways [16]. Our previous paper showed that autophagy-related gene expression was induced by 1% glucose in wild-type (WT) *Arabidopsis* Col-0, but not in *rgs1* mutant plants [3], indicating that RGS1 promoted autophagy-related gene mRNA is induced by glucose. It has been proposed that Heterotrimeric G protein complex and regulator of G-protein signaling 1 (AtRGS1) is a glucose receptor or co-receptor in G-protein-mediated glucose sensing [17,18,19]. AtRGS1 is a GTPase activating protein (GAP) that acts on the G subunit of the heterotrimeric G protein complex. The molecule of *Arabidopsis* RGS1 includes an *N*-terminal seven transmembrane (7TM) domain fused with a cytoplasmic RGS domain [17,18,19]. AtRGS1 is a member of a glucose-sensing complex in *Arabidopsis thaliana* [20]. D-Glucose regulates AtRGS1 activity by interacting with G protein alpha subunit [17,19]. When AtRGS1 is associated with GPA1, it turns off signaling by reducing the level of GTP-bound form of the G protein or through its GAP activity. Removal of bound glucose from the G protein complex activates the G-protein signaling pathway as the *Arabidopsis* G subunit spontaneously binds GTP without requiring a guanine nucleotide exchange factor, such as the G-protein coupled receptor in animals [21,22].

The turnover of RGS1 protein in heterotrimeric G-protein signaling is important for numerous processes, including cell cycle regulation and cellular signaling. Study of the autophagy role in RGS1 homeostasis and glucose-induced metabolism will uncover the signaling/metabolic pathways involved in these processes and whether RGS1 is involved in autophagosome formation. In this study, we showed that nutrient starvation increases proteasome-independent AtRGS1 degradation, which is correlated with increased autophagic flux. Co-localization of RGS1 and autophagosomes was observed in *Arabidopsis* root cells and BY-2 cells. Glucose treatment induced autophagosome formation at early stages. RGS1 and its truncations can interact with the autophagosome marker ATG8a. This study demonstrates for the first time that the autophagosome participates in RGS1 metabolism in plants and that ATG8a co-localizes and interacts with RGS1 and its 7TM.

## 2. Results

### 2.1. Glucose Decreased Autophagic Flux and Increased Formation of Autophagosomes in Plant Cells

We first measured the formation of autophagosomes by induction with glucose [3]. As shown in (Figure 1A,D) cells supplied with the nutrients contained fewer autophagosomes, and the majority of the green fluorescent protein (GFP)-tagged autophagic marker ATG8a [23,24] were distributed evenly in the cytoplasm. Concanamycin A (CA) is an inhibitor of the vacuolar type H^+^-ATPase that has been used to inhibit degradation by reducing the activity of vacuoles to stabilize autophagic cargos [25,26], as in the observed autophagosome addition. Nutrient starvation increased the amount of autophagosomes in root cells (Figure 1B,D). Glucose increased the formation of autophagosomes in plant cells (Figure 1C,D), suggesting that starvation and glucose treatment induced autophagosome formation.

In addition, we measured the level of autophagic flux in plants as this determines the entire process of autophagy, including autophagosome formation, fusion with vacuoles and consequent breakdown, and the liberation of amino acids and fatty acids to the cytosol [27,28,29]. GFP-ATG8/light chain 3 (LC3) is the standard marker for autophagic flux, which is monitored by the release of its GFP tag [30,31]. An increase in autophagosomes in the cell is due to suppression of autophagosome maturation or increase in autophagic flux [27,28]. Consequently, quantitating autophagic flux is critical when analyzing the levels of key signaling molecules, such as AtRGS1 in this case. Therefore, we quantified the delivered amount of autophagy reporter, rather than relying solely on the autophagosome number.

Autophagic flux, as measured by the release of GFP from GFP-ATG8a, was approximately 10-fold higher after starvation (Figure 1E). The 26S-proteasome inhibitor MG-115 that is known to increase autophagy [32] and the protein synthesis inhibitor cycloheximide that partially inhibits autophagy [33], were used to validate that this change was due to autophagy. Indeed, MG-115 increased flux and cycloheximide reduced flux (Figure 1E). When cells were re-fed with glucose, autophagic flux returned to a lower level within 3 h (Figure 1F).

### 2.2. RGS1 Promoted the Production of Autophagic Flux and Autophagosomes

To determine the relationship between RGS1 and autophagosome and autophagic flux, we performed screening by crossing *rgs1* and GFP-ATG8a and obtained homozygous seeds containing GFP-ATG8a in *rgs1*. Autophagic flux was measured by Western blotting using GFP antibodies in *rgs1* (GFP-ATG8a staining) transgenic *Arabidopsis*. As shown in (Figure 2A,B) in GFP-ATG8a, compared with the control, the content of free GFP was lower in 1% glucose treatment for 6 and 12 h. In *rgs1* (GFP-ATG8a staining), 1% glucose treatment was performed separately for 0, 6 and 12 h, and no significant change was observed in free GFP expression, suggesting that RGS1 promotes the production of autophagic flux. We further investigated the root cells of GFP-ATG8a transgenic seedlings to observe whether autophagosomes were induced by 1% glucose treatment for 0.5 h in *Arabidopsis thaliana*. Therefore, the green punctate structures were examined using laser confocal scanning microscopy, before and after glucose treatment. Glucose led to dramatic changes, showing a significant increase in autophagosomes (Figure 2C,E). However, in 1% glucose treatment for 0.5 h a small amount of autophagosomes were induced in *rgs1* (GFP-ATG8a staining) (Figure 2D,E). These results indicate that RGS1 promotes the formation of autophagosomes.

### 2.3. RGS1 Degradation is Correlated with Increased Autophagic Flux Independent of the Proteasome

Interestingly, we observed that the change in autophagic flux was due to degradation and re-feeding of RGS1 protein resulted in the opposite event. After starvation of sucrose for 2 h, the steady-state level of AtRGS1 decreased, and autophagic flux increased (Figure 3A–C). Upon addition of glucose, the level of AtRGS1 recovered and autophagic flux returned to the baseline. Thus, alteration of RGS1 steady state levels was induced by nutrient starvation and addition of 1% glucose. The starvation-induced decrease of total AtRGS1 occurred in the presence of proteasome inhibitor MG-115 (Figure 3D,E), suggesting that the decrease in AtRGS1 results from autophagy and not by proteasome. As MG-115 itself induces autophagy, the addition of this inhibitor further decreased the steady-state level of AtRGS1 in comparison with nutrient starved conditions. To verify whether AtRGS1 vesicles can fuse with autophagosomes containing ATG8a, we designed the following experiments to show the co-localization of RGS1 and autophagosomes.

### 2.4. Co-Localization of RGS1 and Autophagosomes

Seedlings of GFP-ATG8a and RGS1-red fluorescent protein (RFP) were observed for colocalization of autophagosomes with RGS1 in *Arabidopsis* root cells. In comparison with the autophagosome formation and AtRGS1 under normal growth conditions (Figure 4A) (Appendix A), after half an hour post-treatment with 6% glucose of cells, autophagosomes and AtRGS1 were found to overlap (Figure 4B) (Appendix A), indicating their co-localization was internalized by glucose induced endocytosis.

BY-2 cells are relatively homogenous and are suitable for studies of autophagy [34,35]. LysoTracker Red (LTR) dye is an effective autophagosome stain used to quantitate autophagy activity [36,37,38], and it can slightly stain *Nicotiana tobaccos* RGS1 (NtRGS1) in cytoplasm especially on the plasma membrane. The results showed that the punctate fluorescent signal of GFP-tagged tobacco RGS1 (NtRGS1-GFP) overlapped with the LTR punctate signals, with an estimate of ~35% co-localization of the two markers (Figure 4C) (Appendix A), whereas starvation increased co-localization to ~70% (Figure 4D) (Appendix A). This finding is consistent with results obtained by Hanamata et al. using the autophagic flux marker yellow fluorescent protein (YFP)-tagged NtATG8a for autophagosome formation [35]. We found that autophagosomes were increased by 0.5 h sucrose treatment after 2 days starvation and that a portion of RGS1-GFP was located in LTR punctate signals (autophagosome), implying that parts of RGS1-GFP were recycled in autophagosomes.

Starvation of BY2 cells increased the NtRGS1-GFP punctate signals and specific punctate structures stained with LTR (Figure 4D). The increased co-localization suggests that starvation drives NtRGS1 into the autophagosome. Our BY-2 cellular experiment showed the significant increase in autophagosome after 3% sucrose treatment (Figure 4E), consistent with induction of autophagosomes in *Arabidopsis* by glucose (Figure 1C). We observed that NtRGS1^1–248^-GFP, NtRGS1^249–413^-GFP, and NtRGS1^249–459^-GFP in BY-2 cells overlapped with LTR punctate signals (Figure 5), suggesting that co-localization of NtRGS1 truncations and autophagosomes occurred by 0.5 h sucrose induction.

FM4-64 dye is widely used in the study of plasma membrane and vesicles; it combines with plasma membrane and endometrial organelles to produce high-intensity fluorescence. This dye is used for the observation of autophagosomes, such as in tobacco cells, under sucrose starvation treatment, the flow of plasma membrane to the autophagosome membrane was observed by FM4-64 dye, and the central vacuole component was autophagosome, indicating that endocytosis and supply from central vacuoles may aid in the formation of autophagosomes [39]. Related studies have shown that accumulation of autophagosomes in *Arabidopsis* root cells and yeast cells can be detected using FM4-64 [40,41]. Here, we observed the confocal of *Arabidopsis thaliana* root cells treated with 3% glucose and stained with FM4-64 dye. The results of the FM4-64 dye encapsulating the yellow fluorescent RGS1-YFP illustrates the close relationship between autophagy and RGS1 (Appendix A).

### 2.5. ATG8a Interacts with Full-Length and Truncated RGS1

Our previous paper showed that ATG2 and ATG5 inhibited AtRGS1 recovery after D-glucose treatment, whereas 1% glucose treatment induced ATG gene expression [3]. Both ATG2 and ATG5 play important roles in autophagosome formation [3,42]. AtRGS1-YFP is located on the plasma membrane in WT and *atg2* and *atg5* mutant plants (Figure 6A,D). After addition of 6% glucose for 0.5 h, autophagy inhibitor 3-methyladenine (3-MA) affected the subcellular localization of AtRGS1-YFP (Figure 6B,E), as shown by an increase in AtRGS1-YFP vesicles. This result suggests the internal location of AtRGS1-YFP in the plasma membrane. In the absence of 3-MA, the number of AtRGS1-YFP vesicle was reduced in *atg2* and *atg5* mutants compared with the WT (Figure 6C,F), implying that sequestration of autophagy-related proteins (at least ATG2 and ATG5) participates in AtRGS1 metabolism. As AtRGS1 endocytosis is a type of stimulation, we would assume that ATG2 and ATG5 might provoke the stimulation of RGS1 endocytosis. To further confirm that the autophagosome marker GFP-ATG8a interacts with RGS1 and its truncations, we carried out bimolecular fluorescence complementation (BiFC) and pull-down experiments.

In the BiFC experiment (Figure 6G), fluorescence was complemented in the cells expressing AtRGS1-cYFP, which was tagged with the autophagy protein nYFP-ATG8a, but not in cells expressing the negative control P31-nYFP. ATG8 is a ubiquitin-like protein involved in cargo recruitment and biogenesis of autophagosomes. Autophagosome size is determined by the amount of ATG8. As ATG8 is selectively enclosed by autophagosomes, its breakdown would allow measurement of the autophagic rate [30].

To verify the interaction of ATG8a with RGS1 and its truncations, we used purified protein His-ATG8a to pull down RGS1 and its truncations, which were expressed in *Nicotiana benthamiana* leaves by *Agrobacterium* infiltration. ATG8a pulled down RGS1 and its truncations including RGS1-GFP, GFP, RGS1^1–283^, RGS1^1–413^, RGS1^1–283,414–459^, RGS1^1–250^, and RGS1^14–250^ (Figure 7), indicating that both RGS1 7TM and RGS1 domains interact with ATG8a. Altogether, our data demonstrate that the autophagosome is closely related to RGS1.

## 3. Discussion

Throughout the eukaryotic kingdom, RGS proteins act as negative regulators in G protein signaling. In animal cells, RGS proteins participate in cellular processes, including cell growth, mitosis, neuron signaling, membrane diffusion, embryo development, and inflammatory and neurodegenerative diseases [20,43,44,45]. In humans and nematodes, RGS proteins are involved in almost all signaling transmission and adjustment processes. AtRGS1 serves as an important sensor for glucose in *Arabidopsis* [19,20]. The study of the relationship between plant cell autophagy and RGS1 is instrumental to our understanding of the role of RGS1 in sugar signaling.

Our study showed the metabolism of RGS1 is negatively correlated with autophagic flux (Figure 3A,C). Autophagic flux and autophagosomes of *rgs1* mutant express all rarely processing at different times. (Figure 2). In addition, we noted that in co-localization of AtRGS1 and ATG8a after glucose-induced (Figure 4B) co-localization of NtRGS1 and autophagosomes (with LTR stain) under normal conditions, starvation treatment and sucrose recovery (Figure 4C–E), the rate of co-localization under starvation treatment was higher than that under normal conditions, that is, sucrose induced more autophagosomes (Figure 4C,E). We further confirmed that ATG8a can interact with RGS1 and its truncations (Figure 7). FM4-64 as a marker of endocytosis is able to trace the formation of autophagosomes, with RGS1-YFP being wrapped by this dye (Appendix A). These results suggest that autophagy may participate in RGS1 degradation during starvation and RGS1 endocytosis by inducement with glucose. Autophagosomes are induced by treatment with glucose (Figure 1C,D) and sucrose (Figure 4E) at the early phase stage. We speculate that exuberant metabolism may be associated with increased autophagosomes. Xiong et al. observed that 30 mM (about 0.54%) glucose treatment inhibited autophagy through inducing activity of protein kinase target of rapamycin (TOR) [46]. RGS1 promotes ATG expression after glucose induction [3]. Our results show that 1–6% glucose treatment in the early phase promoted autophagosome formation (Figure 1C) and negatively regulated autophagic flux in *Arabidopsis* (Figure 2B). The 3% sucrose treatment induced the production of autophagosomes in BY2 cells after starvation (Figure 4E). We used mannitol as a control treatment and observed that autophagosomes treated by 3% glucose were induced by glucose rather than osmotic stress (Appendix A).

## 4. Materials and Methods

### 4.1. Plant Materials and Growth Conditions

All *Arabidopsis* lines were of the Columbia ecotype. Seed surface was sterilized with 70% ethanol for 10 min, 95% ethanol for 10 min, and finally washed with water. Seeds were vernalized at 4 °C for 2 days in ½ X Murashige and Skoog (MS) liquid medium supplemented with 1% sucrose. A total of 100 *Arabidopsis* seed were grown in 100 mL liquid medium in a 250 mL flask with rotary shaking at 140 RPM under dim continuous light (40 µE m^−2^·s^−1^) at 23 °C for 7 days.

After sterile rinsing thrice with water and starvation using ½ MS lacking sucrose for 2 h, the seedlings were transferred to ½ MS with 100 mM MG115 or 70 µM CHX in continuous dim light for 6 h, whereas the same medium with 1% glucose was used as control. All experiments were performed at least thrice. For immunoblot analyses, seedlings were harvested and flash frozen in liquid nitrogen.

WT *Nicotiana benthamiana* plants were used for BiFC and pull-down experiments. Plants were grown at 23 °C and 70% relative humidity under a 16 h light/8 h dark cycle for 1–1.5 months before infiltration. After infiltration, plants were kept under the same growth conditions.

### 4.2. Arabidopsis Thaliana Mutants and Transgenic Lines

*atg2* (SALK_076727)*,* and *atg5* (SALK_020601) were obtained from the *Arabidopsis* Biological Resource Center. AtRGS1 (encoding amino acids 1–459) was subcloned to pEarleyGate205 (C-terminal TAP). The 35S::AtRGS1-YFP construct was transformed into *Agrobacterium* EHA105, which was then used to transform WT, *atg2*, and *atg5* by the floral-dip method [47], homozygous lines of transgenic plants were used in this study.

The coding regions of RGS1 were amplified with TaKaRa Ex Taq DNA Polymerase (Fisher Scientific, R001A) using specific primers containing Gateway attB sites and then cloned into pDONR207 (Appendix A) entry vector using the BP Clonase II (Life Technologies) to create RGS1 entry clones. After verification by sequencing, each clone was mobilized using the LR Clonase II (Life Technologies) into the Gateway destination vector pK7RWG2 (Appendix A). The primers used for pENTR clones in autophagy-related gene. The 35S::AtRGS1-RFP construct were transformed into Agrobacterium EHA105, which was then used to transform the transgenic *Arabidopsis* expressing GFP-ATG8a (Dr. Faqiang Li providing) by the floral-dip method [47]; homozygous line RGS1-RFP/GFP-ATG8a of transgenic plants was used in this study (Appendix A).

Homozygous *rgs1* (GFP-ATG8a staining) was successfully crossed through crossing *rgs1* and GFP-ATG8a. In general, first cross *rgs1* and GFP-ATG8a and the next generation were screened in Basta-resistant medium to confirm the cross success (*rgs1* is a hybrid, confocal observation was performed to select the strong seedlings and transfer them to the soil, and obtain seeds). Then, the materials collected from each plant were grown on 10 µg/mL Basta-resistant medium and the seedlings on all the grown plates were selected. Confocal observation confirmed a relatively strong GFP fluorescence, the seedlings were transferred to the soil, and each strain was confirmed to be a *rgs1* mutant (Appendix A). In theory, the seedlings were homozygous *rgs1* (stain GFP-ATG8a), and the next generation confirmed *rgs1* (stain GFP-ATG8a) as homozygous.

### 4.3. AtRGS1-YFP Internalization Analysis

Fluorescence quantification for AtRGS1-YFP internalization was performed as described by Urano et al. [48] and Fu et al. [49]. WT, *atg2,* and *atg5* seedlings (7 days old) were treated with 0% or 6% D-glucose (*w*/*v*) for 30 min. Root epidermal cells located 2–4 mm below the cotyledon were imaged (Z stacks obtained) using a Zeiss LSM710 confocal laser scanning microscope equipped with a 20 × Plan-NeoFluor numerical aperture (N.A. = 0.5) objective and a 40 × C-Apochromat (N.A. = 1.2) water immersion objective. YFP fluorescence was excited by a 514 nm argon laser and detected at 526–569 nm by a photomultiplier detector. At least 10 sets of images from seven seedlings were obtained for internalization quantification analysis by ImageJ software.

### 4.4. BiFC

BiFC was pe work of Klopffleisch et al. [50]. The coding regions of ATG8a were amplified with TaKaRa Ex Taq DNA Polymerase (Fisher Scientific, RR001A, Invitrogen, Waltham, MA, USA) using specific primers containing Gateway attB sites and then cloned into the pDONR207 entry vector (Appendix A) using the BP Clonase II (rformed as described in the Life Technologies, Invitrogen) to create ATG8a entry clones. After verification by sequencing, each clone was mobilized using the LR Clonase II (Life Technologies) into the Gateway destination vector pCL112_JO (Appendix A) for BiFC. The primers used for pENTR clones in autophagy-related gene ATG8a are listed as follows: GGGGACAAGTTTGTACAAAAAAGCAGGCTTCATGATCTTTGCTTGCTTGAAATT and GGGGACCACTTTGTACAAGAAACTGGGTCTCAAGCAACGGTAAGAGATCCAAAAGT. The open reading frame of AtRGS1 in BiFC vectors was as previously described by Grigston et al. [51].

*Agrobacterium tumefaciens* strain EHA105 was cultured on Luria Bertani (LB) medium for 2 days, and a single colony was inoculated into 5 mL LB medium supplemented with the appropriate antibiotics (spectinomycin) and grown at 28 °C in a shaker for 48 h. The culture was transferred to the infiltration buffer with 10 mM 2-(*N*-morpholine)-ethanesulfonic acid (MES; pH 5.6) and 40 µM acetosyringone (1:100 ratio, *v*/*v*) for growth at 28 °C for 16 h. When growth reached A_600_ = 3.0, the bacteria were spun down gently (3200g, 10 min), and the pellets were resuspended in 10 mM MgCl_2_ at a final A_600_ = 1.5 (A_600_ = 1 for p19). A final 150 µM acetosyringone was added and the bacteria were kept at room temperature for at least 4 h without shaking.

Split nYFP- and cYFP-tagged protein pairs (nYFP-ATG8a and RGS1-cYFP; P31-nYFP and RGS1-cYFP), p19 (gene silencing suppressor), and mitochondrial RFP marker (Mt-rk, an internal transformation control) were co-expressed in 4–5-week-old Nicotiana benthamiana leaves by Agrobacterium *tumifaciens*-mediated infection. Leaf infiltration was conducted by fully depressing a 1 mL suspension with a syringe onto the surface of leaves. Infiltrated leaves exhibited a water-soaked appearance. Images were captured 3 days after inoculation.

Leaf disks were obtained from infiltration sites and imaged by confocal microscopy as previously described [48]. Tobacco leaf epidermal cells were imaged using a Zeiss LSM710 confocal laser scanning microscope equipped with an Apochromat 40× water immersion objective (N.A. = 1.2). YFP fluorescence was excited by a 514 nm argon laser and detected at 526–569 nm by a photomultiplier detector, and Mt-rk and RFP fluorescence was excited by a 543nm HeNe laser and detected at 565–621 nm.

### 4.5. Transformation of BY-2 Cells

BY-2 cells (*Nicotiana tabacum* “Bright Yellow”) were cultured in modified MS medium supplemented with 3% (*w*/*v*) sucrose, 1 µg/mL thiamine-HCl, 0.2 µg/mL 2,4-D, 100 µg/mL myo-inositol, and 200 µg/mL KH_2_PO_4_, with a final pH 5.8 adjusted by KOH. Cell lines were cultured in either liquid MS medium with continuous shaking, or in the form of *calli* on MS media solidified with 0.8% (*w*/*v*) agar in the dark at 26 °C. Suspensions were sub-cultured every 7 days by transferring 1.5 mL culture into 30 mL fresh MS medium; *calli* were sub-cultured every 3–4 weeks. NtRGS1 was expressed under 35S promoter in our constructs. The tobacco homolog of AtRGS1, the NtRGS1 gene was amplified from BY-2 cDNA and cloned into pDrive cloning vector with 12 bp of Kozak sequence to achieve seamLess expression. The fragment was subsequently cloned into the pGreen binary vector using BamHI and HindIII to create C-terminal fusion with enhanced GFP. Partial digestion was required for this procedure.

Transformation of BY-2 cells was performed using *Agrobacterium* tumefaciens carrying the binary vector pCP60 harboring NtRGS1^1–459^-GFP, NtRGS1^1–248^-GFP, and NtRGS1^249–413^-GFP, NtRGS1^249–459^-GFP [34,52]. The primers were as follows: GGAGAATAAATTATGGCAGCTTG and GCAGTTTTGAATCATGACTATGG GFP-tagged Nicotiana tabacum NtRGS1^1–459^-GFP, NtRGS1^1–248^, NtRGS1^249–459^, and NtRGS1^249–413^, see Domain structure of the RGS1 protein in Appendix A.

Exponential cell suspension (3–4 days after sub-culturing) was filtered and resuspended in 30 mL of fresh MS medium. Acetosyringone (15 mL, 40 mM) was added to the suspension and thoroughly mixed by pipetting. Total of 3 mL *Agrobacterium* suspension was then added to the cell suspension and cultivated for 3 days in the dark at 26 °C. The cells were washed with 300 mL 3% (*w*/*v*) sucrose and 100 mL MS medium supplemented with 100 mg/L cefotaxime. Finally, the cells were resuspended in 2–3 mL liquid MS medium containing 100 mg/L cefotaxime and 50 mg/L kanamycin and cultured in a Petri dish for 3–4 weeks in the dark at 26 °C. The *calli* cultures were transferred onto fresh MS medium with the same antibiotics. Grown cells (3 d sub-culturing) were centrifuged at 5000g for 10 min, rinsed thrice with sucrose-free BY-2 medium and subsequently starved in MS medium for 2 days. BY-2 cells grown in the MS medium supplemented with 3% sucrose served as control.

### 4.6. Co-Localization of RGS1 and Autophagosome

WT, *atg2,* and *atg5* RGS1-RFP/GFP-ATG8a seedlings (7 days old) were treated with 0% or 6% D-glucose (*w*/*v*) for 30 min. Root epidermal cells located 2–4 mm below the cotyledon were imaged (Z stacks obtained) using a Zeiss LSM710 confocal laser scanning microscope equipped with a 20 × Plan-NeoFluor (N.A. = 0.5) objective and a 40 × C-Apochromat (N.A. = 1.20) water immersion objective. GFP fluorescence was excited by a 488 nm argon laser and detected at 505–550 nm by a photomultiplier detector, whereas RFP fluorescence was excited by a 543 nm HeNe laser and detected at 565–621 nm.

CA (1 µM) and LTR (1 µM) Red (Invitrogen) were added into BY-2 media for 12 and 3 h, respectively, before confocal imaging. GFP fluorescence was excited by a 488 nm argon laser and detected at 505–550 nm by a photomultiplier detector, and LTR fluorescence was excited by a 543 nm HeNe laser and detected at 565–621 nm. At least 10 sets of images were obtained thrice BY-2 cells for quantification analysis. GFP or LTR punctae were counted per cell according to 10 sets of images field of vision.

### 4.7. Pull Down Assays

pENTR clones in ATG8a were mobilized using the LR Clonase II (Life Technologies) into the Gateway destination vector pDEST17 (Appendix A) for His-ATG8a. His-ATG8a was expressed in *E. coli* at 13 °C for 2 days after isopropyl β-D-1-thiogalactopyranoside induction. Purificating ATG8a protein was used with Ni-NTA agarose (QIAGEN). For the pull down assay, *Agrobacterium tumefaciens* RGS1-GFP, RGS1^1–283^-GFP, GFP, RGS1^1–413^-GFP, RGS1^1–283,414–459^-GFP, RGS1^1–250^-GFP, and RGS1^14–250^-GFP [53] were transiently expressed in *N. benthamiana* for 3 days. Transient expression in tobacco was determined as previously described [54], except for the infiltration buffer (10 mM MgCl_2_, 10 mM MES, and 150 µM acetosyringone). The harvested sample was ground in liquid nitrogen and suspended in a buffer (100 mM Tris-HCl pH7.5), 150 mM NaCl, 1 mM ethylenediaminetetraacetic acid (EDTA), 0.5% Nonidet P-40, plus protease inhibitor cocktail (100X, 100 mM phenylmethylsulfonyl fluoride (PMSF), 1 µg/mL aprotinin, 1 µg/mL leupeptin, and 1 µg/mL pepstatin) and phosphatase inhibitor cocktail (100X, 1 mM Na_3_VO_4_, 1 mM NaF). After centrifugation at 12,000× *g* for 12 min, the supernatants were set for pull down assay.

ATG8a protein was incubated with the above protein extracts of RGS1 (and its truncations)-GFP or GFP and was expressed for 2 h at 4 °C. The beads were washed thrice with the extraction buffer, and proteins were analyzed by sodium dodecyl sulfate polyacrylamide gel electrophoresis (SDS-PAGE). The RGS1 (and its truncations)-GFP and GFP were detected by Western blotting using the GFP antibody (JL-8 Monoclonal Antibody, Fisher, Invitrogen, Waltham, MA, USA).

### 4.8. Immunoblot Analyses

The total amount of AtRGS1-TAP (AtRGS1 encoding amino acids 1–459) was subcloned to pEarleyGate205 (C-terminal TAP) for quantitative analysis by immunoblots. AtRGS1 tagged with TAP [48] was detected with peroxidase anti-peroxidase soluble complex (PAP). RGS1-TAP seedlings were used in all cases to quantitate AtRGS1. Plants expressing the autophagy marker GFP-ATG8a were used to quantitate autophagic flux using an antibody directed against GFP (Roche). Plant-actin (E12-053) was obtained from Enogene® Biotech (New York, NY, United States).

RGS1-TAP and GFP-ATG8a seedling extracts were suspended in buffer A (50 mM Tris-HCl pH 7.5), 150 mM NaCl, 5 mM EDTA, 0.2% Triton X-100, 0.2% Nonidet P-40, and 1:100 PMSF (stock 60 mM and 2% ABS-14, 1% β-mercaptoethanol) were centrifuged twice at 12,000 rpm for 15 min. Protein concentration was measured with BIO-RAD Protein Assay. β-actin levels were analyzed as loading controls for different conditions [28].

### 4.9. Confocal Microscopy

The images were obtained in Zeiss LSM 710 confocal laser-scanning microscope (Carl Zeiss Microscopy GmbH, Jena, Germany) and analyzed with Aim Image Browser Image Processing software.

For the observation of seedlings’ roots autophagosomes. Autophagosomes labeled by GFP-ATG8a in root cells of GFP-ATG8a plants with incubation in 1 µM concanamycin A (CA) for 12 h. For each D-glucose condition, at least two (typically three or four) seedlings were imaged. GFP fluorescence was excited by a 488 nm argon laser and detected at 505–550 nm by a photomultiplier detector [3,28]. At least 10 sets of images from seven seedlings were obtained for quantification analysis (autophagosomes were counted by the number of GFP points at every root cell in 10 images).

For observation of co-localization of AtRGS1 and autophagosomes with RFP-RGS/GFP-ATG8a seedlings, for each D-glucose condition, at least two (typically three or four) seedlings were imaged. Red fluorescence was excited by a 488 nm argon laser and detected at 505–550 nm by a photomultiplier detector, and GFP fluorescence was excited by a 543 nm HeNe laser and detected at 565–621 nm.

For observations of co-localization of NtRGS1 (its truncations) and autophagosomes with BY-2, please see the above “Co-localization of RGS1 and autophagosome”.

RGS1-YFP seedlings (7 day old) were treated with 3% D-glucose (*w*/*v*) and FM4-64 for 2 h. Root cells located 2–4 mm below the cotyledon were imaged with a Zeiss LSM880 using a META system (LCSM; Carl-Zeiss, Jena, Germany) Confocal microscopy with excitation relationship between autophagy and RGS1 at 561 nm (a multi-Ar ion laser) emission at 603–680 nm was used to detect the autophagsomes. YFP signals were excited by a 514-nm argon laser and its emission was detected at 518–588 nm by a photomultiplier detector. ImageJ plugin was used for image quantification. Digital Images were captured with a 40× oil immersion objective and analyzed with ZEN software (Carl Zeiss).

### 4.10. Statistical Analysis

All tests were repeated at least three times and the results were analyzed by GraphPad Prism 5. Statistical analyses were performed using Student’s t test (*p* < 0.05, *p* < 0.01, and *p* < 0.001). Quantification of the bands on western blots was performed using image analyzing software, Image J [48,49]. Each value was the mean ± S.D. of three independent replicates.

## Figures and Tables

**Figure 1 ijms-20-04190-f001:**
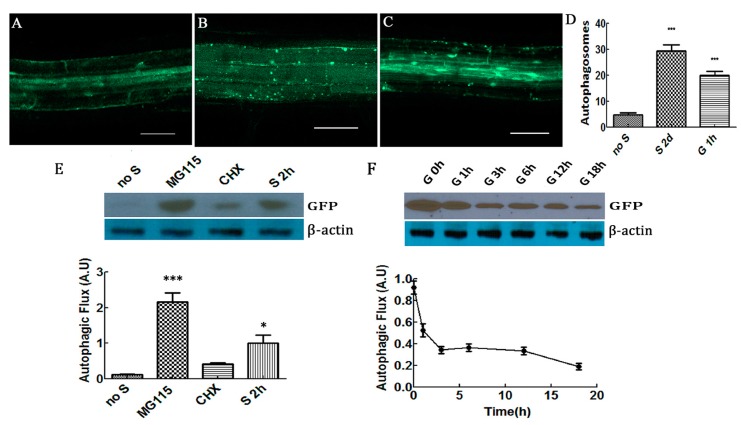
Proteasome-independent autophagosomes were induced by starvation and glucose. (**A**) Seven-day-old seedlings expressing GFP-ATG8a grown in ½ MS medium containing 1% sucrose for 7 days. (**B**) Seven-day-old seedlings grown as described for panel A except for use of sucrose-free medium for 2 days. (**C**) Seven-day-old seedlings grown as described for panel A remove sucrose being transferred to media with added 2% glucose for 1 h. Fluorescence from GFP-ATG8a visualized with a 488 nm excitation and 505–550 nm emission settings. A, B, C autophagosomes labeled by GFP-ATG8a in root cells incubation within 1 µM CA for 12 h. Scale bars = 50 µm. (**D**) Quantification of autophagosomes in root cells of four seedlings: “no S” means no starvation (**A**); “S 2d”, starvation for 2 days (**B**); “G 1h”, 2% glucose treatment for 1 h (**C**). (**E**) Seven-day-old seedlings grown in liquid ½ MS, and 1% sucrose under dim, continuous light. “Starved” indicates the seedlings treated with ½ MS media without sucrose for 2 h. Shown above is a typical Western blot probed with antiserum against GFP. Quantification of bands from replicate Western blots is shown below. MG-115, starvation plus 100 mM MG115 for 2 h CHX, starvation plus 70 µM CHX for 2 h; “S 2 h”, starvation with no sucrose for 2 h as control. (**F**) Glucose indicated by “G” was added at the indicated times. β-actin was used as loading controls. Each value was the mean ± S.D. of four independent replicates. Asterisks indicate significant differences (* *p* < 0.05 or *** *p* < 0.001).

**Figure 2 ijms-20-04190-f002:**
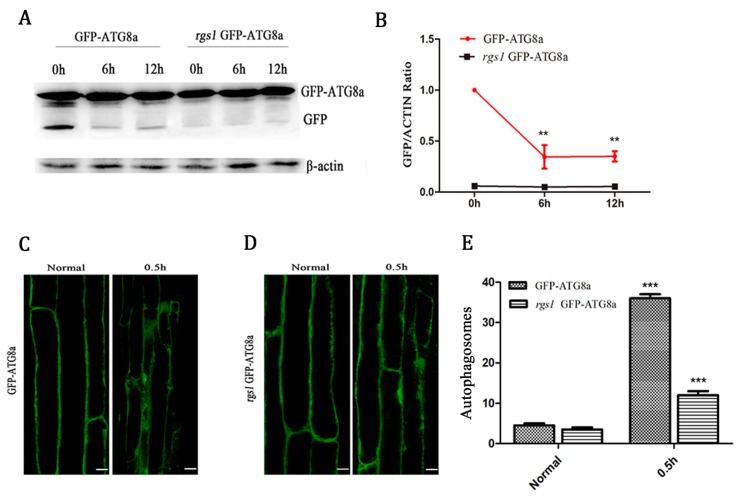
RGS1 promoted the production of autophagosomes and autophagic flux. Quantification of autophagic flux and GFP-ATG8a and *rgs1* (GFP-ATG8a staining) seedlings. Seven-day-old seedlings of GFP-ATG8a treated by liquid MS medium without sugar following stimulation by 1% glucose for 0, 6, and 12 h. (**A**) Equal amounts of protein extracted from the seedlings were used in SDS-PAGE, followed by Western blotting with anti-GFP and anti-β-actin antibodies. (**B**) Quantification of changes in free GFP normalized with the expression of β-actin. Asterisks indicate significant differences compared to starved seedlings treated with 1% glucose for 0 h ** *p* < 0.01. Error bar represent S.D. obtained from three independent replicates. (**C**) Observation of autophagosomes. Autophagosomes labeled by GFP-ATG8a and *rgs1* (GFP-ATG8a staining), in roots of GFP-ATG8a plants incubated with 1 µM CA for 12 h. Normal seedlings and GFP-ATG8a in seedlings treated by 1% glucose for 0.5 h in GFP- ATG8a. (**D**) Normal seedlings and 1% glucose seedlings for 0.5 h in *rgs1* (GFP-ATG8a staining) seedlings treated by 1% glucose for 0.5 h. (**E**) Quantification of GFP-ATG8a and *rgs1* (GFP-ATG8a staining), labeled autophagosomes per root cell at the indicated times were used to calculate autophagic activity. Mean and S.D. values were calculated from roots of six seedlings per time point. Results from three parallel experiments were used for quantification. Asterisks indicate significant differences of the starved seedlings treated with glucose from normal ones. Scale bars = 10 µm (** *p* < 0.01 or *** *p* < 0.001).

**Figure 3 ijms-20-04190-f003:**
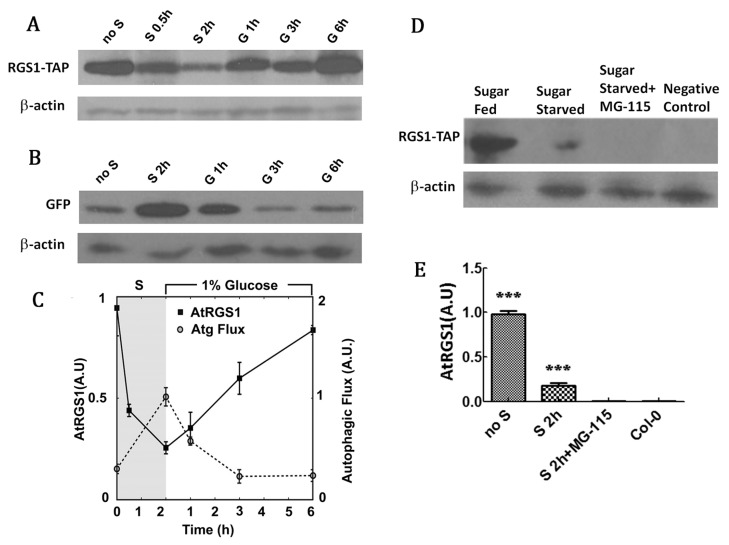
Levels of RGS1 and autophagic flux were observed during starvation and recovery. (**A**) TAP tagged AtRGS1. Seven-day-old seedlings were cultured in ½ X MS 1% sucrose under dim, continuous light, and then transferred to media with no sucrose (no S, Starved) and treated with 1% glucose for 1, 3, and 6 h (G 1 h, G 3 h, and G 6 h, respectively); Western blot was probed with PAP antibody. (**B**) Autophagic flux. Same as for A except the Western blot was probed with antiserum directed against GFP. (**C**) Quantitation of levels of AtRGS1-TAP and autophagic flux under starvation (S) or 1% glucose for various hours. (**D**) Seedlings were treated as described in A except with MG115 (100 mM) treatment in liquid ½ X MS or no MG115 for 2 h. (**E**) Quantification of changes in D. Each value was the mean ± S.D. of four independent replicates. β-actin was used as loading controls. Asterisks indicate significant differences (*** *p*<0.001).

**Figure 4 ijms-20-04190-f004:**
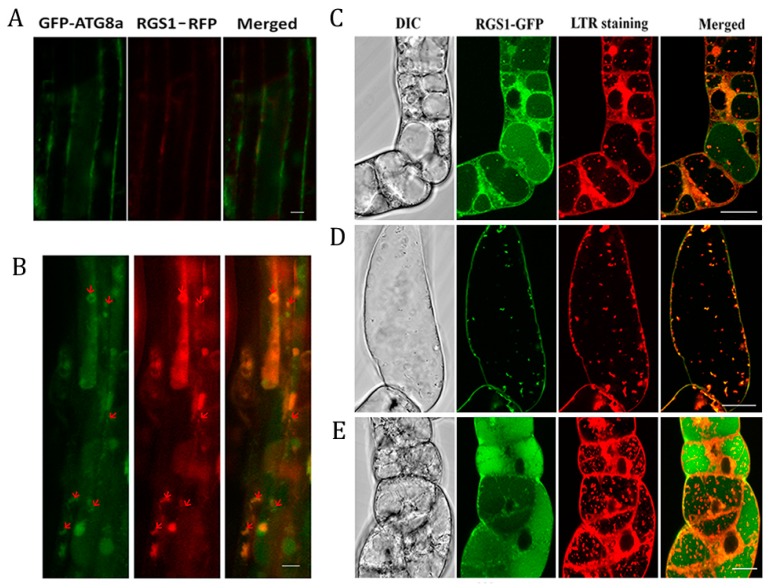
Co-localization of RGS1 and autophagosomes was detected in *Arabidopsis* root cells and BY-2 cells. (**A**) Absence of endocytosis of AtRGS1 with RGS1-RFP/GFP-ATG8a seedlings under normal growth conditions (½ MS medium containing 1% sucrose for 7 days) was observed. (**B**) Endocytosis of AtRGS1-RFP remove sucrose being transferred to media with added 2% glucose for 1 h by GFP-ATG8a, and showing co-localization of GFP-ATG8a and AtRGS1-RFP. (**A**) and (**B**) Scale bar = 10 µm. GFP-tagged *Nicotiana tabacum NtRGS1* (**C**) under MS 3% sucrose treatment for 5 days, and added with 1µM CA 12 h (control). (**D**) RGS1 with MS 3% sucrose treatment for 3 days, starvation for 2 days, and added with 0.5 µM CA for 12 h (starvation). (**E**) RGS1 with MS 3% sucrose treatment for 3 days, starvation (MS, no sucrose) for 2 days, and 3% sucrose restoration for 0.5 h, showing numerous autophagosomes. Differential contrast microscopy (DIC), merged image of green (or yellow) and red. *NtRGS1-GFP* BY-2 cells were standardly cultured for 3 days, and then supplemented with 3% sucrose (control) or without sucrose (starvation). Con A (1 µM) and LTR stain were added for confocal imaging. LTR fluorescence appeared red, and RGS1-GFP appeared green. In the merged images, the overlap of LTR and RGS1 fluorescence appeared yellow. (**C**) (**D**) (**E**) Scale bar = 20 µm. Autophagosomes stained with LTR.

**Figure 5 ijms-20-04190-f005:**
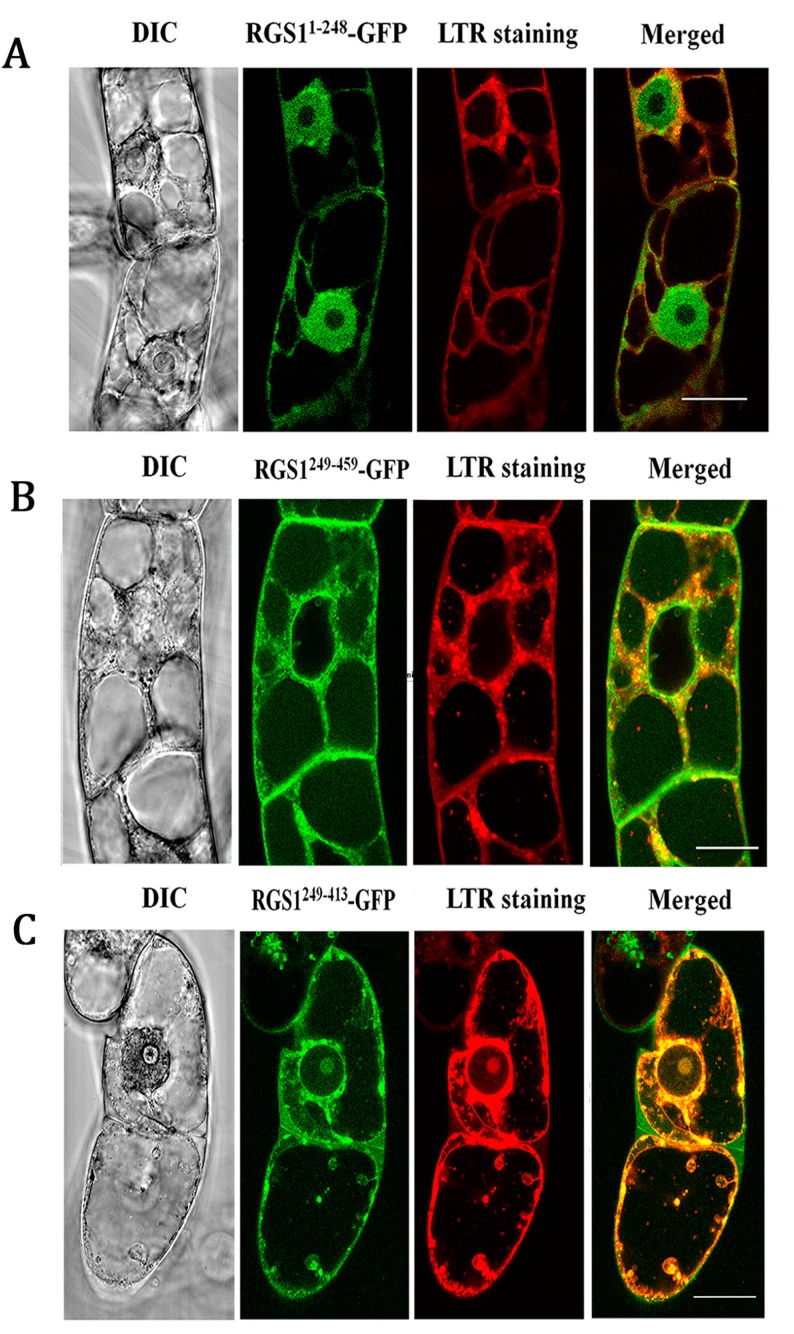
Co-localization of NtRGS^11–248^-GFP, NtRGS^1249–459^-GFP, NtRGS^1249–413^-GFP and autophagosomes was observed in BY-2 cells. (**A**) BY-2 cells expressing GFP-tagged *Nicotiana tabacum NtRGS1*^1–248^, (**B**) NtRGS1^249–459^, and (**C**) NtRGS1^249–4`13^**.** BY-2 cells were grown under normal conditions (MS 3% sucrose) for 7 days, with 1µM (final concentration) CA added for 12 h, and 3% sucrose was re-added for 0.5 h. DIC, LTR fluorescence appeared red, RGS1-GFP gives green, and the merged images of LTR and RGS1 fluorescence presented a yellow color. Scale bar = 20 µm. Four independent replicates.

**Figure 6 ijms-20-04190-f006:**
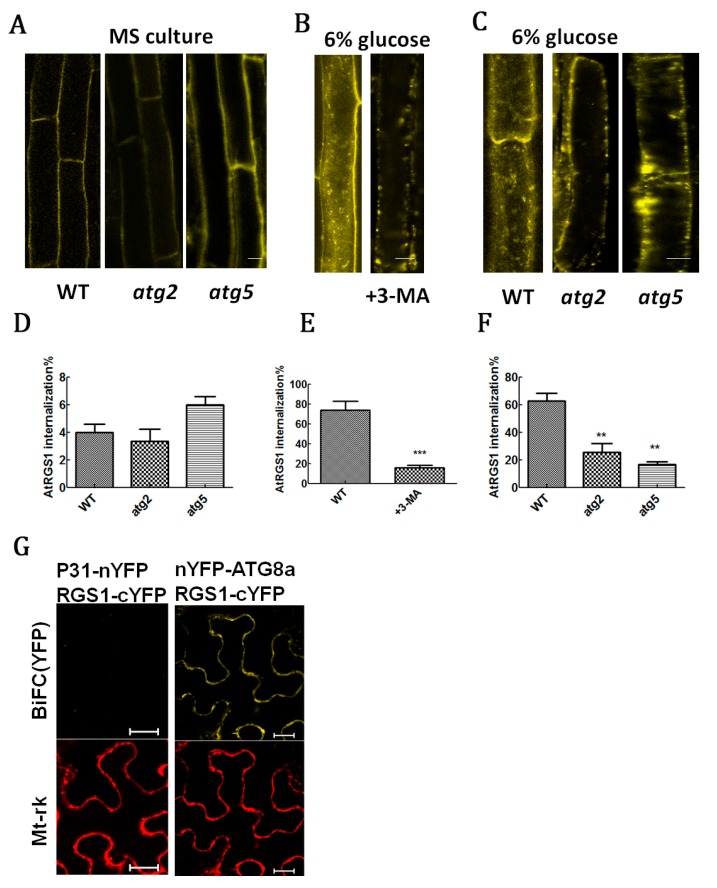
AtRGS1-YFP endocytosis was limited in *atg2* and *atg5*, AtRGS1 interacted with ATG8a in vivo. (**A**) Location of WT, *atg2*, and *atg5* AtRGS1-YFP in the plasma membrane under normal growth conditions (½ MS medium containing 1% sucrose for 7 days). (**B**) Endocytosis of AtRGS1-YFP remove sucrose being transferred to media with added 6% glucose and was inhibited by autophagy inhibitor 3-MA. (**C**) Comparison with the endocytosis of AtRGS1-YFP in Columbia ecotype induced by removing sucrose being transferred to media with added 6% glucose for 0.5 h, which was inhibited in *atg2* and *atg5*. Image obtained with confocal microscopy for AtRGS1 internalization amount (see Materials and Methods). (**D**) Quantification of changes in A. (**E**) Quantification of changes in B. (**F**) Quantification of changes in C. (**G**) BiFC was used to test the physical interaction between test proteins. Tobacco pavement cells were co-transformed with the indicated BiFC pairs plus an internal positive transformation control Mt-rk. Transformation was performed as described in Methods. Images were captured on the third day of cell growth and the presented data are the lowest detectable level of expression to avoid ectopic complementation of fluorescence. The RGS1-YFP band is indicated. Scale bar = 20 µm. Each value is the mean ± S.D. of four independent replicates. Asterisks indicate significant differences (** *p*<0.01 or *** *p*<0.001).

**Figure 7 ijms-20-04190-f007:**
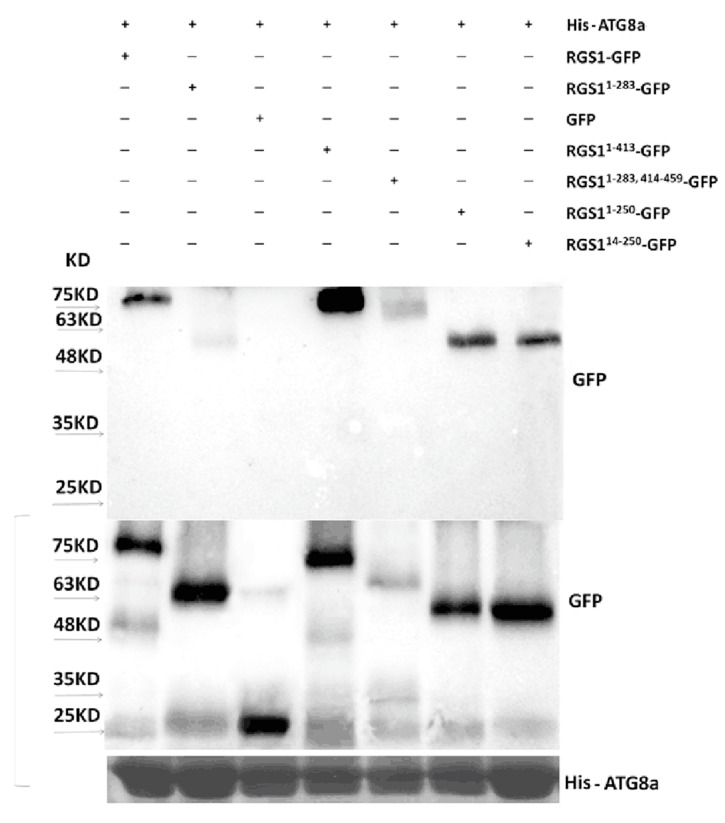
Pull down assay of RGS1 and its truncations by His-tagged ATG8a was performed. Proteins of RGS1-GFP, RGS1^1–283^-GFP, GFP, RGS1^1–413^-GFP, RGS1^1–283,414–459^-GFP, RGS1^1–250^-GFP, RGS1^14–250^-GFP, were purified by His-ATG8a on Ni-NTA agarose column and detected by Western blotting using the GFP antibody (see Materials and Methods).

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
