# Peer review of "Correlation of Autophagosome Formation with Degradation and Endocytosis Arabidopsis Regulator of G-Protein Signaling (RGS1) through ATG8a"

_ijms, 2019, doi:10.3390/ijms20174190_

Round 1
Reviewer 1 Report
The paper is well written, scientifically sound. Figures are clear and high quality.
Minor comments:
Line 20 Change "that ATG8a of autophagosome marker interacts" to "that autophagosome marker ATG8a interacts".
Change reference 1 to a more recent one, like Limanaqi et al. 2019 Cell Clearing Systems Bridging Neuro-Immunity and Synaptic Plasticity. Int J Mol Sci. 20(9):2197 or Ji & Kwon 2017 Crosstalk and Interplay between the Ubiquitin-Proteasome System and Autophagy. Mol Cells. 40(7):441-449.
Line 32 Change "engulfed with membrane" to "engulfed with a double membrane".
Line 33 Change lysosome to lysosomal.
Line 303 Change "1%-6%" to "1-6%".
Panel C of supplementary Figure S1 is extremely dark, hardly anything is visible in it.
In supplementary Figure S6 numbers of the basepair standard are listed irregularly. They should be arranged to the right. It is disturbing that the samples are in reverse order. The figure should be flipped over vertically.
Reviewer 2 Report
The authors explore the regulation of AtRGS1 endocytosis and transport to autophagosomes by glucose and sucrose. They propose that the degradation of AtRGS1, a main glucose sensor, is accelerated by starvation. This seems counter-intuitive: inactive sensor is removed, but activated is not. The data suggest that ATG8a interacts with SAtRGS1 and facilitates its endocytosis and degradation.
Unfortunately, the results section is rather muddled. It needs to be re-written and streamlined. A cartoon illustrating proposed mechanism in the discussion would be most helpful.
What is the physiological relevance of 1-6% glucose? Do these concentrations ever occur in vivo? Is GFP-tagged AtRGS1 still functional as an RGS protein? The reduction in RGS by starvation would increase G protein activity. Is there evidence for that in cells expressing endogenous (untagged and therefore fully functional) proteins? 7. Which blot is input and which is pulled-down material? Some editing is needed. E.g. line 178 “found to be overlapped” should be “found to overlap”; lines 281-2, “G protein-coupled signaling” should be “G protein signaling”; line 300, what do the authors mean by “exuberant metabolism”; etc.Author Response
Please see the attachment

Reviewer 3 Report
This work focuses on the autophagy caused by the starvation and explores the role of G protein signalling in this process particularly via RGS1 protein. The experimental part seems to be well done, but I would suggest to make sure the data are clearly presented using arrows, e.g. in Figure 1, it is not entirely clear what was quantified. The figure legends seems to be to brief in some cases, missing words, e.g. light, dim "conditions", perhaps some language correction would help in some cases.
Overall, the text should be improved to be more explanatory (see e.g. Front Plant Sci. 2019; 10: 280 as an example) putting the topics in more general context. In the introduction, there are many details but more general introduction what is the connection between the starvation, autophagy, glucose, G signalling is missing. The same occurs in the experimental design. The set up conditions are not clearly explained - starvation vs. added glucose conditions, etc.
I suggest to make a graphical summary of obtained data.
Also, the discussion is rather extended data summary and lacks the clear line. It would be helpful to put the new observations into more general context. Aain, the graphical model should help to clarify the message.
Lines 291-293: the sentence is too long, lacking the clear meaning. Please clarify.
Round 2
Reviewer 2 Report
The authors adequately responded to some comments, but not to all. In particular, they did not explain biological relevance of 6% glucose (reference to another paper is hardly an explanation) and did not change Fig. 7 to make it clearer to the reader.
Some added statements seem self-contradictory: “Our study showed for the metabolism of RGS1 is negatively correlated with autophagic flux (Figure 3A, C), RGS1 promotes the production of autophagosomes and autophagic flux (Figure 4B)” (lines 292-293). Isn’t RGS1 degraded by autophagy? The authors need to clarify what they mean by “autophagic flux”.
In many cases the language needs to be improved: e.g., “use of 1/2 MS medium containing 1% sucrose were grown for 7 days” and similar additions to figure legends, etc.
Author Response
Response to Reviewer 2 Comments
Comments to the Author
Manuscript ID: ijms-549341
Title: Correlation of autophagosome formation with degradation and endocytosis Arabidopsis regulator of G-Protein Signaling (RGS1) through ATG8a
Response: Thanks a lot for your positive and instructive comments. We completely agree your comments, and have made some modifications for our manuscript according your opinions.
Point 1: The authors adequately responded to some comments, but not to all. In particular, they did not explain biological relevance of 6% glucose (reference to another paper is hardly an explanation)
Response 1:The reason for using this concentration is that Professor Chen ever used the concentration in Alan Jones laboratory to observe the endocytic situation of RGS1 in a short time.
Point 2: did not change Fig. 7 to make it clearer to the reader.
Response 2: We are very sorry for our unclear description,we have changed figure legends and materials and methods.
As follows:
ATG8a pulled down RGS1 and its truncations including RGS1-GFP, GFP, RGS11-283, RGS11-413, RGS11-283,414-459, RGS11-250, and RGS114-250 (Figure 7), indicating that both RGS1 7TM and RGS1 domains interact with ATG8a. Altogether, our data demonstrate that the autophagosome is closely related to RGS1.
Figure 7. Pull down assay of RGS1 and its truncations by His-tagged ATG8a was performed. Proteins of RGS1-GFP, RGS11-283-GFP, GFP, RGS11-413-GFP, RGS11-283,414-459-GFP, RGS11-250-GFP, RGS114-250-GFP, were purified by His-ATG8a on Ni-NTA agarose column and detected by Western blotting using the GFP antibody (see Materials and Methods).
For the pull down assay, Agrobacterium tumefaciens RGS1-GFP, RGS11-283-GFP, GFP, RGS11-413-GFP, RGS11-283,414-459-GFP, RGS11-250-GFP, and RGS114-250-GFP [1] were transiently expressed in N. benthamiana for 3 days.
References :[1]Hu, G.; Suo, Y.; Huang, J. A crucial role of the RGS domain in trans-Golgi network export of AtRGS1 in the protein secretory pathway. Molecular plant 2013, 6, 1933–1944
Point 3: Some added statements seem self-contradictory: “Our study showed for the metabolism of RGS1 is negatively correlated with autophagic flux (Figure 3A, C), (lines 292-293). Isn’t RGS1 degraded by autophagy? The authors need to clarify what they mean by “autophagic flux”.
Response 3:(Figure 3A, C)This result of the figure is concluded by treating 1% glucose after 2 h of starvation, which indicates that the expression of RGS1 is significantly reduced in the case of starvation (no sugar) for 2 h, and returns to the original level 6 h after the administration of 1% glucose. The change of autophagic flux is exactly the opposite of RGS1, so we speculate that autophagy may be involved in the metabolic process of RGS1.
RGS1 promotes the production of autophagosomes and autophagic flux (Figure 2)”
This sentence is written according to our experimental results. It can be seen that the autophagic flux and autophagosomes in the rgs1 mutant are all rare under any circumstances, and the relation of autophagic flux and autophagosomes is being studied in our laboratory. We found that the relationship in this case is complicated, and it is not consistent between the formation of autophagic flux and autophagosomes.
we have changed (lines 292-294) to “ Our study showed the metabolism of RGS1 is negatively correlated with autophagic flux (Figure 3A, C). Autophagic flux and autophagosomes of rgs1 mutant express all rarely processing at different times (Figure 2).
The authors need to clarify what they mean by “autophagic flux”.
We describe the definition of autophagic flux in (lines 87-92 )of the article.
As follows: In addition, we measured the level of autophagic flux in plants as it determines the entire process of autophagy, including autophagosome formation, fusion with vacuoles and consequent breakdown, and liberation of amino acids and fatty acids to the cytosol [27–29]. GFP-ATG8/light chain 3 (LC3) is the standard marker for autophagic flux, which is monitored by the release of its GFP tag [30,31]. An increase in autophagosomes in the cell is due to suppression of autophagosome maturation or increase in autophagic flux [27,28]. Consequently, quantitating autophagic flux is critical when analyzing the levels of key signaling molecules such as AtRGS1 in this case. Therefore, we quantified the delivered amount of autophagy reporter, rather than relying solely on autophagosome number.
Point 4: In many cases the language needs to be improved: e.g., “use of 1/2 MS medium containing 1% sucrose were grown for 7 days” and similar additions to figure legends, etc.
Response 4: we have changed “use of 1/2 MS medium containing 1% sucrose were grown for 7 days” to “½ MS medium containing 1% sucrose for 7 days”
